# Identification of Candidate Genes Involved in Fruit Ripening and Crispness Retention Through Transcriptome Analyses of a ‘Honeycrisp’ Population

**DOI:** 10.3390/plants9101335

**Published:** 2020-10-10

**Authors:** Hsueh-Yuan Chang, Cindy B. S. Tong

**Affiliations:** Department of Horticultural Science, University of Minnesota, Saint Paul, MN 55108, USA; c-tong@umn.edu

**Keywords:** fruit ripening, cell wall, xyloglucan endotransglucosylase/hydrolase (MdXTH), RNA-Seq, nCounter^®^

## Abstract

Crispness retention is a postharvest trait that fruit of the ’Honeycrisp’ apple and some of its progeny possess. To investigate the molecular mechanisms of crispness retention, progeny individuals derived from a ’Honeycrisp’ × MN1764 population with fruit that either retain crispness (named “Retain”), lose crispness (named “Lose”), or that are not crisp at harvest (named “Non-crisp”) were selected for transcriptomic comparisons. Differentially expressed genes (DEGs) were identified using RNA-Seq, and the expression levels of the DEGs were validated using nCounter^®^. Functional annotation of the DEGs revealed distinct ripening behaviors between fruit of the “Retain” and “Non-crisp” individuals, characterized by opposing expression patterns of auxin- and ethylene-related genes. However, both types of genes were highly expressed in the fruit of “Lose” individuals and ’Honeycrisp’, which led to the potential involvements of genes encoding auxin-conjugating enzyme (GH3), ubiquitin ligase (ETO), and jasmonate O-methyltransferase (JMT) in regulating fruit ripening. Cell wall-related genes also differentiated the phenotypic groups; greater numbers of cell wall synthesis genes were highly expressed in fruit of the “Retain” individuals and ’Honeycrisp’ when compared with “Non-crisp” individuals and MN1764. On the other hand, the phenotypic differences between fruit of the “Retain” and “Lose” individuals could be attributed to the functioning of fewer cell wall-modifying genes. A cell wall-modifying gene, *MdXTH*, was consistently identified as differentially expressed in those fruit over two years in this study, so is a major candidate for crispness retention.

## 1. Introduction

’Honeycrisp’ (*Malus domestica* Borkh.), known for its exceptional fruit crispness, is an apple cultivar developed by the University of Minnesota [1]. Since its release in 1991, ’Honeycrisp’ has become the fourth most produced apple cultivar in Washington state, the major USA apple producer, with the highest increase in growing acreage over years [2]. A crisp apple fruit texture is desired by consumers [3,4]. Unlike most commercial apple cultivars with fruit that lose crispness during ripening and postharvest storage, ’Honeycrisp’ fruit retain their crisp texture for at least six-months of cold storage [5].

Ethylene, a plant hormone essential for the ripening of climacteric fruit [6], could be related to ’Honeycrisp’ fruit crispness retention. Typically, a burst of ethylene production can be observed in apple fruit at the onset of fruit ripening, which triggers a series of physiological changes, including losses in two related traits, firmness (compression force) and crispness (fracturability) [7]. However, ethylene concentrations in ’Honeycrisp’ fruit are relatively low and stable during ripening, especially compared with ’McIntosh’, a variety that exhibits the usual climacteric ethylene production and rapid fruit softening [8]. Low ethylene production is a feature of apple varieties other than ’Honeycrisp’ with fruit that retain higher firmness (less softening) during postharvest storage [9,10]. *MdACS1* and *MdACO1*, the major genes responsible for climacteric ethylene production in apple fruit, are also correlated with postharvest changes in fruit firmness [9]. In particular, *MdACS1*, with two allelotypes, *MdACS1-1* and *MdACS1-2* associated with different ethylene production rates and fruit softening patterns, has been used as a molecular marker for long shelf life apples [11,12].

The cell wall, responsible for the crisp texture of apple fruit [13], is another focus of crispness retention research. Tong et al. [5] showed that ’Honeycrisp’ fruit were able to maintain integrity of the cell wall after long-term storage without obvious degradation of the middle lamella, a region important for cell-to-cell adhesion and fruit crispness [13,14]. Pectin is the main component of middle lamella, and thus lack of middle lamella degradation of ’Honeycrisp’ fruit can be related to genes involved in pectin degradation. As pectin is the most structurally complex plant cell wall polysaccharide [15], the degradation of pectin is accomplished by the synergistic functions of various genes. In apples, the expression patterns of several pectolytic genes, such as *Mdα-AF1, Mdα-AF2, Mdα-AF3*, *Mdβ-GAL2*, *MdPG1*, and *MdPME1*, have been correlated with fruit softening in specific varieties [16,17,18,19,20]. Among these genes, *MdPG1* may have major effects on crispness retention, since its gene expression was consistently low in ’Honeycrisp’ fruit during ripening and storage [8,21]. In addition, the correlation between *MdPG1* and crispness retention was supported by its low enzyme activity in ’Honeycrisp’ fruit [22].

In addition to pectin, cellulose and hemicellulose are other major structural polysaccharides in the plant cell wall. Non-pectolytic genes, such as *MdEXPs* and *MdXTHs*, involved in reorganizing the cellulose–hemicellulose network, could also affect the texture of apple fruit [23,24,25], but their roles in ’Honeycrisp’ crispness retention have not been clearly established. In a genetic study, Harb et al. [8] showed that *MdEXP2* and *MdXTH2* had lower gene expression in ’Honeycrisp’ fruit during fruit ripening compared with ’McIntosh’ fruit, while *MdXTH10* exhibited an opposite expression pattern. Trujillo et al. [26], on the other hand, found that *MdEXP2* was not strongly related to crispness retention by examining the gene expression and effects of allelotype in fruit of several apple varieties and ’Honeycrisp’ progeny individuals. So far, there has been no additional study to validate the functions of *MdXTH*s in ’Honeycrisp’ fruit, and thus the *MdXTH* gene family is a potential target for further analysis.

Toward a more comprehensive understanding of the molecular mechanisms behind apple fruit crispness retention, the transcriptomes of fruit from a population derived from ’Honeycrisp’ × MN1764 were studied. MN1764 is an advanced breeding selection in the apple breeding program at the University of Minnesota. Unlike ’Honeycrisp’, the fruit of MN1764 were less crisp at harvest and lose crispness during storage [27]. This cross between ’Honeycrisp’ × MN1764 generated progeny individuals with fruit that differ in the ability to retain postharvest crispness [27]. RNA-Seq, as a high-throughput sequencing-based method [28], was first applied to identify differentially expressed genes (DEGs) associated with the crispness retention of the ’Honeycrisp’ × MN1764 population. The expression patterns of the DEGs were then validated using nCounter^®^, a mid-throughput hybridization-based method [29], by including second-year samples. With the uses of a genetically-related apple population and transcriptomic approaches, candidate genes for crispness retention can be more reliably identified.

## 2. Results

### 2.1. Phenotype and Transcriptome Variations Among the Individuals

Fruit of the selected progeny individuals and parents were distinct in puncture force (PF), force linear distance (FLD), and acoustic pressure (AUX) (Figure 1). Crisp fruit were characterized with higher force (FLD and PF) and acoustic (AUX) values compared with the non-crisp fruit. As reported in our previous study [27], the average PF, FLD, and AUX of this breeding population at harvest were 5.1 kg, 2144, and 72.6 dB. Using these numbers as the thresholds, fruit with all three instrumental measures higher than these numbers were considered as crisp. Among the six individuals with crisp fruit at harvest, three of them retained their force and acoustic values after 8-week cold storage while the others did not. Based on the instrumental measures, three phenotypes were distinguishable: 1) individuals with fruit that retained crispness (CL136, EF129, and EF138), 2) individuals with fruit that lost crispness (CF117, CL156, and EF117), and 3) individuals with fruits that were not crisp at harvest (CL121 and CL124). The terms “Retain”, “Lose”, and “Non-crisp” were used to designate the three phenotypes. Based on the instrumental measures, ’Honeycrisp’ fruit retained, while MN1764 fruit lost, crispness.

Differences among the “Retain”, “Lose”, and “Non-crisp” individuals were observed at the transcriptomic level. A multidimensional scaling (MDS) plot (Figure 2) showed that samples with the same phenotypes had similar gene expression patterns. A separation was detected between freshly-harvested and stored samples. Freshly-harvested “Non-crisp” samples clustered apart from the “Retain” and “Lose” individuals, but after storage, the individuals were not as widely separated as at harvest. ’Honeycrisp’ samples clustered with other crisp individuals, while MN1764 replicates were spread between crisp and non-crisp samples.

There were 4129, 2030, and 6870 DEGs identified in comparisons between ’Honeycrisp’ and MN1764, the “Retain” and “Lose” individuals, and the “Retain” and “Non-crisp” individuals, respectively (Figure 3). A total number of 107 genes were found in all three comparisons. Of these, 67 genes were commonly expressed in fruit of ’Honeycrisp’ and the “Retain” individuals, and therefore potentially related to the trait of retaining crispness (Figure 3a). Additionally, 40 genes were commonly expressed in fruit of MN1764 and the “Non-crisp” and “Lose” individuals and are therefore possibly related to losing crispness (Figure 3b).

### 2.2. Functional Annotation of Differentially Expressed Genes

The larger transcriptomic variations between fruit of the “Non-crisp” and ”Retain” individuals also were reflected in the higher number of enriched Gene Ontology (GO) terms (Figure 4a). Most differentially expressed genes (DEGs) highly expressed in fruit of the “Retain” but not in the “Non-crisp” individuals were associated with the oxidation-reduction process (GO:0055114), signal transduction (GO:0007165) and peptidyl-tyrosine modification (GO:0018212). For the enriched GO terms with more specific biological functions, the response to auxin (GO:0009733), response to light stimulus (GO:0009416), and carbohydrate transport (GO:0008643) represented about 2% of the differentially expressed genes (DEGs). On the other hand, DEGs highly expressed in fruit of the “Lose” but not in the “Retain” individuals were mainly enriched in the defense response (GO:0006952), rRNA processing (GO:0006365), response to karrikin (GO:0080167), and fruit ripening (GO:0009835).

A different group of GO terms were identified between fruit of the “Retain” and “Lose” individuals at harvest compared with the “Retain” versus “Non-crisp” individuals (Figure 4b). The most enriched GO terms were the RNA modification (GO:0009451), and microtubule severing (GO:0051013) for the “Retain” fruits, and the DNA rewinding (GO:0036292) and oligosaccharide catabolic process (GO:0009313) for the “Lose” fruits. Functional similarity of the DEGs were observed after storage. Two of the significantly enriched GO terms, the oxidation-reduction process (GO:0055114) and response to karrikin (GO:0080167), for fruit of the “Retain” and “Lose” individuals respectively, were also identified between fruit of the “Retain” and “Non-crisp” individuals.

There were only three GO terms identified between ’Honeycrisp’ and MN1764 fruit (Figure 4c). The DEGs highly expressed in ’Honeycrisp’ and MN1764 fruit were consistently enriched in signal transduction (GO:0007165) and carbohydrate transport (GO:0008643) at harvest and storage, and the number of genes within each term were similar. These two GO terms were also identified in comparison of fruit of the “Retain” and “Non-crisp” individuals (Figure 4a).

### 2.3. The Expression Patterns of Auxin- and Ethylene-Related Genes

Genes involved in the auxin signaling pathway, due to their potential roles in fruit ripening, were analyzed to extend the results of the functional annotation where genes associated with response to auxin (GO:0009733) were enriched (Figure 5a). In the comparison between fruit of the “Retain” and “Non-crisp” individuals, most auxin-related genes, including *ARF*, *AUX/IAA*, *SAUR*, and *GH3*, were highly expressed in the “Retain” fruits, especially at harvest (Figure 5). Higher expression of auxin-related genes was also found in ’Honeycrisp’ fruit compared with MN1764 fruit, but some of them were differentially expressed after storage (Figure 5). In contrast, fewer auxin-related genes were differentially expressed between fruit of the “Retain” and “Lose” individuals, and most of them were highly expressed in the “Lose” fruits (Figure 5a–c) except for the *GH3* groups, which were highly expressed in the “Retain” fruits (Figure 5d). *IAA/AUX* (*MD02G1057200*) and *GH3* (*MD05G1092300*) were the only auxin-related genes identified in all three comparisons. Additionally, three *SAUR* genes, MD02G1133900, MD04G1082600, and MD14G1152100, were differentially expressed in two of three comparisons.

For ethylene-related genes, the major difference was the higher number of *1-aminocyclopropane 1-carboxylate synthases* (*ACSs*) genes expressed in fruit of the “Non-crisp” and “Lose” individuals compared to that of the “Retain” individuals (Figure 6a). Among them was *MdACS1* (*MD15G1302200*), the *ACS* responsible for climacteric ethylene production in apple fruit (Figure 6a). However, the number of *ACSs* differentially expressed between the parents were similar and *MdACS1* was highly expressed in ’Honeycrisp’ fruit (Figure 6a). Two types of genes displayed specific expression patterns in the “Non-crisp” fruit: low expression of two *ethylene overproducer genes* (*ETOs)* (Figure 6a) and high expression of one *ethylene response sensor* (*ERS*) gene (Figure 6b). The other two types of ethylene-related genes, *1-aminocyclopropane 1-carboxylate oxidases* (*ACOs*) and *ethylene response factors* (*ERFs)*, were differentially expressed across comparisons, but there was no consistent pattern in terms of the number of DEGs identified in specific individuals (Figure 6a and c). For example, about half of the differentially expressed *ACOs* were highly expressed in “Non-crisp” fruit, while another half were highly expressed in “Retain” fruits. *MdACO1* (*MD10G1328100*), another gene involved in climacteric ethylene production, was not differentially expressed among the individuals in this study.

Two *ACS* genes, *MdACS6* (*MD06G1090600*) and *MdACS3a* (*MD15G1203500*), were identified in all three comparisons (Figure 6a). *MdACS6* was highly expressed in fruit of MN1764 and the “Lose”, and “Non-crisp” individuals, while *MdACS3a* was highly expressed in ’Honeycrisp’ fruit and those of the “Retain” individuals. Three *ERF* genes, *ERF1B-like* (*MD13G1213100*), *MdERF3* (*MD14G1226300*), and *SHN1* (*MD17G1209000*) were differentially expressed in two of the three comparisons (Figure 6c).

### 2.4. The Expression Patterns of Cell Wall-Related Genes

Major cell wall-related genes associated with the texture of apple fruit were studied (Table 1). Based on the total number of DEGs, cell wall-related genes were more highly expressed in fruit of the “Retain” individuals compared with the “Non-crisp” individuals (Table 1a). At harvest, cellulose synthase, galacturonosyltransferase (*GAUT*), pectin methylesterase (*PME*), and xyloglucan endotransglucosylase/hydrolase (*XTH*) were more highly expressed in fruit of the “Retain” individuals compared to “Non-crisp”, while α-arabinofuranosidases (*α-AFs*) were highly expressed in fruit of the “Non-crisp” individuals (Table 1a). After storage, most *cellulose synthases* and *GAUTs* maintained higher expression in fruit of the ’Retain’ than ’Non-crisp’ individuals (Table 1a). Similar patterns of gene expression were detected between ’Honeycrisp’ and MN1764. There was higher expression of *cellulose synthases* and *GAUTs* in ’Honeycrisp’ fruit and *α-AF* in MN1764 fruit, but these patterns were observed only after storage and not at harvest (Table 1c).

In the comparison between fruit of the “Retain” and “Lose” individuals, expression patterns of the cell wall-genes were not as clear as in the comparison between those of the “Retain” and “Non-crisp” individuals (Table 1b). At harvest, most *XTHs* were identified in fruit of the “Retain” individuals but not in those of the “Lose” individuals, while β-galactosidase (*β-GAL*), and cellulose synthase genes were mostly identified in the “Lose”, but not “Retain”, individual’s fruit. After storage, one *GAUT* was identified in fruit of the “Retain”, but not “Lose”, individuals, while expansin genes and *PGs* were mainly identified in the fruit of the “Lose”, but not “Retain”, individuals.

### 2.5. RNA-Seq Results Validation Using nCounter^®^ Technology and qRT-PCR

To validate the RNA-Seq results, nCounter^®^ data were generated from the same RNA samples used for RNA-Seq analyses, as well as samples harvested in 2019. Results depicted in Appendix A showed that gene expression measured by RNA-Seq (in FPKM) and nCounter^®^ (in gene count) were correlated (r = 0.8). Higher correlations were obtained for genes with high expression levels and inconsistencies occurred mostly for genes with low expression levels.

qRT-PCR was performed to further probe the gene expression discrepancies between RNA-Seq and nCounter^®^ results. Appendix A shows that RNA-Seq and nCounter^®^ both failed to detect gene expression in some cases, in contrast with qRT-PCR results. For MD02G1057200 (*auxin-responsive protein*), MD10G1315100 (*XTH*), and MD11G1230200 (unknown gene), qRT-PCR results agreed with the RNA-Seq data, but for MD05G1098700 (*AMP-dependent synthetase*) and MD12G1164900 (pentatricopeptide repeat-containing protein), qRT-PCR results agreed with nCounter^®^ data. One of the tested genes encoding a CASP-like protein 1F2 (MD14G1150200) was not correctly detected using RNA-Seq and nCounter^®^ technology. Expressions of two cell wall-modifying genes, *XTH* (*MD16G1091200*) and *PG* (*MD10G1179100*), were well correlated among the three methods.

Among the 47 tested genes using nCounter^®^ (Appendix A), 26 genes were differentially expressed in the parent and progeny samples (primary candidate gene) and 14 genes were differentially expressed only in the parent or progeny samples (secondary candidate gene) (Table 2). The other 7 genes identified as differentially expressed by RNA-Seq were not differentially expressed over both years in which fruit were harvested. These candidates included those involved in cell wall modification, as well as plant hormone signaling and biosynthesis. The cell wall-modifying genes were *Md*α-*AF3* (*MD16G1158300*), *GAUT* (*MD17G1141200*), *MdPG1* (*MD10G1179100*), and *XTHs* (*MD10G1315100*, *MD16G1091200*), and the hormone-related genes were AUX/IAA (*MD02G1057200*), *GH3.1* (*MD05G1092300*), *ACSs* (*MD06G1090600*, *MD15G1203500*), and *jasmonate methyltransferase* (*JMT*, *MD15G1023600*).

## 3. Discussion

### 3.1. Fruit Ripening and Crispness Retention

Different ripening characteristics could be one factor causing crisp and non-crisp phenotypes at harvest. Based on the transcriptomic data, fruit of the “Non-crisp” individuals exhibited the typical climacteric ripening processes of apple fruit, in which ethylene plays a crucial role. Genes highly expressed in the “Non-crisp” fruit were enriched in expression of fruit ripening (Figure 4a) genes, and *ACS* genes encoding the rate-limiting enzyme of ethylene biosynthesis [30] were the major DEGs in this GO term (Figure 6a). One member of the *ACS* family, *MdACS1* (*MD15G1302200*), which is essential for increased ethylene production during apple fruit ripening [11], was highly expressed in the “Non-crisp” fruit. Another important *ACS* in apple fruit, *MdACS3a* (*MD15G1203500*), associated with relatively stable ethylene production before climacteric ripening [31], was highly expressed in the “Retain” fruit. *MdACS6* (*MD06G1090600*) also displayed greater expression in “Non-crisp”, “Lose”, and MN1764 fruit compared to “Retain” and ’Honeycrisp’ fruit. This gene was reported by An et al. [32] and Zhao et al. [33] to be up-regulated during fruit ripening or following ethylene production due to salt stress.

The differential-expression patterns of several auxin-related genes further indicated that the fruit of the “Retain” and “Non-crisp” individuals are under different physiological control mechanisms (Figure 5). Auxins have been reported to both promote and inhibit fruit ripening. For example, exogenous auxin application induces ethylene production in ’Golden Delicious’ apple fruit before harvest [34]. In contrast, the delayed-ripening phenotype of the transgenic MADS8/9-suppressed apple was attributed to the maintenance of high auxin concentrations in the fruit [35]. Although the two reports are seemingly contradictory, they both hypothesize that auxin is critical at the onset of fruit ripening. A fruit ripening model for apple established by Busatto et al. [36] demonstrates that after fruit enter advanced ripening stages, auxin-related genes are downregulated along with increased expression of ethylene-related genes. Thus, the higher expression of auxin-related genes observed in this study suggests that fruit of the “Retain” individuals were harvested at relatively early ripening stages.

*Gretchen Hagen 3* (*GH3*) genes, which encode auxin-conjugating enzymes, could be alternative ripening indicators for fruit of the “Retain” individuals wherein ethylene-related genes did not show a differential response. It has been proposed that GH3 initiates the ripening processes of apple fruit due to its capacity to convert biologically active auxin to inactive amino acid [37]. Among the *GH3* genes identified in this study (Figure 5d, Table 2a), *GH3.1* (*MD05G1092300*) was also observed to increase in expression in ’Royal Gala’ apples during fruit maturation and ripening [38], and homologous genes have been correlated with the ripening of other fruits, such as grape (*Vitis vinifera* L.) [39] and pungent pepper (*Capsicum chinense* L.) [40].

The softening phenotype of fruit of “Lose” individuals, characterized as being crisp at harvest and non-crisp after storage, could result from interactions between auxin and ethylene. Ethylene appeared to be involved in ripening of the “Lose” fruits, since several *ACS* genes were highly expressed in fruit of the “Lose” individuals compared with those of the “Retain” individuals (Figure 6a). *MdACS1* (*MD15G1302200*) was not differentially expressed at harvest but during storage of the “Lose” fruits, which correlated with the period during which they started to lose crispness. The delayed-ripening characteristic of the “Lose” fruits might be attributed to auxin. In the comparison between the “Retain” and “Lose” fruits, auxin related-genes, except for the *GH3* group, were more highly expressed in fruit of the “Lose” individuals. It is possible that the lower expression of *GH3* genes caused higher concentrations of active auxin in fruit of the “Lose” individuals, which in turn inhibited climacteric ethylene biosynthesis and fruit ripening.

Defense response genes also differentiated the postharvest ripening statuses of the individuals. Fruit ripening and defense responses are closely connected. For example, Shi et al. [41] found that proteins differentially expressed in stored ’Golden Delicious’ fruit mostly belonged to stress and defense response categories. Zheng et al. [42] also reported increases in defense response proteins in ’Golden Delicious’ fruit during ripening and after ethylene treatment. In this study, most of the DEGs between “Retain” and “Non-crisp” fruits were enriched in the GO term “defense response”. Three of the DEGs, including *chitinase* (*MD01G1213100*), *glucan endo-1,3-β-glucosidase* (*MD11G1189000*), and *jasmonate methyltransferase* (*JMT*, *MD15G1023600*), showed consistently higher expression in fruit of the “Lose” compared with the “Retain” individuals (Table 2b). Chitinases and glucan endo-1,3-β-glucosidases are fungi and bacteria cell wall-degrading enzymes [43], while JMTs are involved in the biosynthesis of methyl jasmonate (MeJA), a plant volatile that regulates defense responses [44]. In addition to roles in defense responses, MeJA is also related to fruit ripening. Exogenous application of MeJA enhanced climacteric ethylene production and softening of ’Golden Delicious’ fruit [45], as well as expression of the transcription factors, MdMYC2 (MD16G1274200) and MdERF3 (MD14G1226300), that regulate ethylene biosynthesis [46]. In this study, *MdMYC2* was not differentially expressed, but *MdERF3* displayed matching expression patterns (Figure 6c) to the *JMT* gene (Table 2b). As a result of its biological function, and the timing of expression, the *JMT* gene is one of the candidate genes associated with fruit ripening of MN1764 fruit and that of the “Lose” individuals.

’Honeycrisp’ fruit has a mechanism of ripening distinct from that of the phenotypically-similar “Retain” individuals based on high expression of *MdACS1* (*MD15G1302200*) (Figure 6a). In apples, there are two *MdACS1* allelotypes, *MdACS1-1* and *MdACS1-2*. *MdACS1-1* is associated with high ethylene production and softer fruit, while *MdACS1-2* is associated with low ethylene production and firmer fruit [12]. The high expression of *MdACS1* in ’Honeycrisp’ fruit corresponds to its heterozygous allelotype, including both *MdACS1-1* and *MdACS1-2*, in contrast to that of the homozygous (*MdACS1-2/2*) MN1764 [26], but is inconsistent with its ripening behavior, including crispness retention. Despite exhibiting high *MdACS1* expression, ’Honeycrisp’ fruit produce low amounts of ethylene during ripening compared with ’McIntosh’ fruit, that show climacteric ripening characteristics [8]. Thus, *MdACS1* expression is unrelated with the crispness retention of ’Honeycrisp’ fruit, but low ethylene production could still be an important factor in its ripening characteristic.

An explanation for low ethylene production in ’Honeycrisp’ fruit is post-transcriptional regulation of *MdACS1*. Among genes involved in ethylene biosynthesis, *ethylene overproducers* (*ETOs*) were DEGs identified in fruit of ’Honeycrisp’ and the “Retain” individuals as having higher expression levels than fruit of MN1764 or “Non-crisp” individuals (Figure 6a). ETOs are ubiquitin ligases, which target type 2 ACSs to the proteasome for degradation [47,48]. The classification of ACSs is based on the C-terminal amino acid sequences, and *MdACS1* belongs to the type 2 ACSs [49]. In Arabidopsis, overexpression of *AtETO1* inhibits *AtACS5* enzyme activity and promotes its degradation [50], but the roles of *ETOs* in fruit ripening have not been reported. Without differential expression of the other ethylene synthesis genes, for example *MdACO1* (*MD10G1328100*), the *ETOs* could be candidates for further elucidating the relationship between ethylene and ripening of ’Honeycrisp’ fruit.

### 3.2. Cell Wall-Related Genes and Crispness Retention

The continuing expression of cell wall synthesis genes during storage could be related to the crispness retention of ’Honeycrisp’ fruit. *Cellulose synthase* and *Galacturonosyltransferase* (*GAUT*) are the two types of cell wall synthesis genes investigated in this study, and there were greater numbers of the two genes highly expressed in fruit of ’Honeycrisp’ and the “Retain” individuals after storage when compared with those of MN1764 and the “Non-crisp” individuals (Table 1a,c). The *cellulose synthase* gene family in plants includes *cellulose synthase* and *cellulose synthase-like* genes, which mediate the synthesis of cellulose and hemicellulose, respectively [51,52]. In climacteric apple fruit, the amount of cellulose and hemicellulose are relatively constant during ripening [53], and *cellulose synthase* and *cellulose synthase-like* genes have not been the main foci to postharvest textural changes. In a recent study by Win et al. [54], degradation of cellulose and hemicellulose, correlating with fruit softening, was observed in fruit of ’Summer King’ and ’Green Ball’ apples during long-term storage. Whether the continuous expression of *cellulose synthase* and *cellulose synthase-like* genes in fruit of ’Honeycrisp’ during storage contributes to crispness retention can be further evaluated in future research by measuring the cellulose and hemicellulose content in ’Honeycrisp’ fruit. From another point of view, the higher expression of *cellulose synthases* during ’Honeycrisp’ fruit storage is an additional indicator showing the slow ripening behaviors of ’Honeycrisp’ fruit, because cell wall synthesis genes were usually active during fruit growth and are down-regulated as fruit mature [55]. Unlike typical climacteric ripening, a lack of clear transition between development and ripening could result in minimal changes in the expression patterns of cell wall genes—a possible mechanism that results in ’Honeycrisp’ fruit crispness retention.

GAUTs, key enzymes in pectin biosynthesis, are potentially involved in crispness retention by acting against the pectin-modifying enzymes that cause fruit softening. One of the *GAUT* genes (*MD17G1141200*) was identified as a candidate favorable for crispness retention (Table 2a), showing consistently higher levels of gene expression in ’Honeycrisp’ than MN1764 fruit. Dheilly et al. [55] showed that this *GAUT* was more active before than after harvest, similar to the expression patterns of most cell wall synthesis genes. The high expression levels of several *GAUTs* in ’Honeycrisp’ fruit at harvest and after storage further emphasized the close relationships between the ripening behaviors of ’Honeycrisp’ fruit and the expression of cell wall-related genes. It should be pointed out that the *GAUT* gene (*MD17G1141200*) was not differentially expressed between fruit of “Retain” and “Lose” individuals (Table 2a), implying the involvement of additional genes in crispness retention.

The expression patterns of cell wall-modifying genes were more consistent than those of cell wall synthesis genes in differentiating fruit of the “Retain” and “Lose” individuals, since the cell wall synthesis genes, *cellulose synthases* and *GAUTs*, were not specifically expressed in fruit of the “Retain” individuals (Table 1b). Several cell wall-modifying genes that have been previously reported to affect postharvest fruit texture were observed in the current study to differentiate crispness phenotypes of fruit of the ’Honeycrisp’ × MN1764 population. Among them, *MdPG1* (*MD10G1179100*) was the most prominent gene associated with losing crispness, as significantly high expression of *MdPG1* was consistently observed in fruit of MN1764, the “Lose”, and the “Non-crisp” individuals (Table 1 and Table 2b). Polygalacturonases (PGs) catalyze the hydrolysis of pectin, and the expression of *MdPG1* during fruit ripening has been correlated with increased pectin solubilization, which leads to softening of apple fruit [20,54,56]. Many studies have illustrated the roles of *MdPG1* through different approaches, including the examination of *MdPG1* in fruit of varieties with different firmness [16] and the study of transgenic apple with suppressed *MdPG1* [57]. In this study, while high expression of *MdPG1* correlated with softening of the non-crisp fruit of the studied population, low or no expression of this gene alone cannot completely explain crispness retention. Despite differential expression, there was also an abundance of *MdPG1* mRNA accumulation in fruit of the “Retain” individuals after storage (Table 2b), which did not result in a significant decrease in fruit crispness, suggesting that PG protein synthesis or activity may differ between fruit with different crispness retention phenotypes.

Other pectin-modifying enzymes, such as pectin methylesterase (PME), α-arabinofuranosidase (α-AF), and β-galactosidase (β-GAL), could have partial effects on fruit crispness retention. The function of PMEs is to remove the methyl groups from pectin [58], while α-AFs and β-GALs are responsible for the cleavage of sugar residues from the side chains of pectin polymers [59]. Together, these enzymes have been hypothesized to facilitate pectin degradation during fruit ripening [20]. *MdPME1* (*MD08G1195600*), *Mdα-AF3* (*MD16G1158300*), and *Mdβ-GAL2* (*MD02G1079200*) were the genes corresponding to the pectin-modifying proteins identified in this study (Table 1). Each of these genes has been previously reported to relate to fruit texture. For example, *MdPME1* in ’Jonagold’ apple is related to its postharvest softening [19], *Mdα-AF3* is highly expressed in individuals with mealy fruit [18], and *Mdβ-Gal2* is associated with the softer texture of ’Fuji’ compared with ’Qinguan’ fruit [60]. However, in our study, the relationships between the three pectin-modifying genes and fruit crispness were not strong. After including samples from a second year in our analyses, the only significant difference observed in our study was the expression levels of *Mdα-AF3* between fruit of the “Retain” and “Lose” individuals (Table 2). As a result of the inconsistencies between the parents and progeny individuals and year-to-year variations, *MdPME1* (*MD08G1195600*), *Mdα-AF3* (*MD16G1158300*), and *Mdβ-GAL2* (*MD02G1079200*) were deemed as low- or non-priority candidate genes.

*MdXTH* (*MD16G1091200*), with consistently higher expression in fruit of ’Honeycrisp’ and the “Retain” individuals over two years (Table 1 and Table 2a), has the characteristics of a good candidate gene underlying crispness retention. Xyloglucan is the most abundant hemicellulose in plant cell walls [61], and xyloglucan endotransglucosylase/hydrolase (XTH) is a key enzyme controlling wall strength and extensibility through its modification of the interactions between cellulose and hemicellulose xyloglucan [62]. There are eleven identified members of *MdXTH*, named *MdXTH1* to *MdXTH11* [63], but the specific *MdXTH* (*MD16G1091200*) identified in this study, with sequence similarity to the *Arabidopsis AtXTH33*, has not been formally named. Although not a focus in previous studies, there are clues suggesting that this *MdXTH* may have a role in regulating fruit crispness. In a study comparing ripening-associated gene expression between ’Golden Delicious’ and ’Fuji’ fruit, a consistently higher expression of this *MdXTH* was observed in ’Fuji’ fruit, which are crisper at harvest, with better crispness retention than ’Golden Delicious’ fruit [64]. In addition, the physical position of *MdXTH* (6.3 Mb) on chromosome 16 is within a QTL region (3.2 to 6.9 Mb) previously identified using five ’Honeycrisp’ populations including ’Honeycrisp’ × MN1764 and associated with apple fruit crispness [65]. Previous reports indicate that different members of *XTH*s gene family could either cause fruit softening [24] or maintain fruit firmness [66]. In the current study, this *MdXTH* was favorable for retaining crispness texture properties. The highest expression of *MdXTH* was at harvest (Table 2a), which indicated that this *MdXTH* was activated before the difference in crispness occurred between fruit of the “Retain” and “Lose” individuals. Based on its expression patterns, previously reported QTL results, and its biological functions, MdXTH could contribute significantly to both at harvest crispness and crispness retention of ’Honeycrisp’ fruit. It is not clear how *MdXTH* is regulated, but regulation may be independent of gene sequence differences, since fruit of the “Retain” and “Lose” individuals shared the same QTL allelotype.

## 4. Conclusions

By comparing the transcriptomes among the phenotypic groups (“Retain”, “Lose”, and “Non-crisp”), we observed that several ethylene- and auxin-related genes associated with fruit ripening were differentially expressed. The ethylene synthesis gene, *MdACS1*, responsible for the climacteric ethylene production of apple fruit was related to the lower crispness of the “Non-crisp” fruit at harvest and the “Lose” fruit after storage. In contrast, the expression of *MdACS1* was lower in the “Retain” fruit, where auxin-conjugating genes (*GH3.1*), rather than ethylene genes could be more important in their ripening. A *JMT*, which was consistently differentially expressed over two years, could also be involved in fruit ripening through a different pathway that eventually enhances ethylene production. ’Honeycrisp’ fruit differed from those of the “Retain” individuals in having higher expression of several *MdACS1* genes, suggesting a possible post-transcriptional regulation of *MdACS1* by ETOs.

The investigation of cell wall genes revealed some number of cell wall synthesis genes, such as *cellulose synthases* and *GAUTs*, that were highly expressed in fruit of the “Retain” individuals and ’Honeycrisp’ compared to those of the ’Non-crisp’ individuals and MN1764. Among the cell wall-modifying genes, the expression patterns of *MdPG1* and *MdXTH* were the most closely correlated with crispness retention. In summary, low expression of *MdACS1*, *MdPG1*, and *JMT* along with high expression of *ETO*, *MdXTH,* and other cell wall synthesis genes were favorable for crispness retention of the “Retain” members of the ’Honeycrisp’ × MN1764 population.

## 5. Materials and Methods

### 5.1. Plant Materials

The cultivar ’Honeycrisp’ and an unnamed breeding selection, MN1764, were the parents of the breeding population studied. ’Honeycrisp’ fruit retain crispness during storage, while MN1764 fruit lose crispness. This breeding population is comprised of 170 progeny individuals, and each individual had 2 to 4 replicated trees. The trees were grown at the University of Minnesota Horticultural Research Center in Victoria, Minnesota. Fruit were harvested from late August to early October 2018 and 2019 when the starch index reached the score of four based on an eight-point starch-iodine index chart [67].

Eight progeny individuals derived from the ’Honeycrisp’ × MN1764 population with distinct crispness traits were further selected for transcriptome comparisons. There were 1) three individuals with fruit that retained crispness through 8-week cold storage (named “Retain” individuals); 2) three individuals producing fruit that lost crispness after two months of cold storage (named “Lose” individuals); and 3) two individuals that had non-crisp fruit at harvest (named “Non-crisp” individuals). Three instrumental measures including puncture force (PF), force linear distance (FLD), and maximum acoustic pressure (AUX), that were highly correlated with sensory crispness of this studied apple population were used as the major indicators of fruit crispness [27,68]. PF was generated using the penetrometer (FT 30, Wagner Instruments, Greenwich, CT) mounted on a drill press and equipped with a FT716 size plunger, while FLD and AUX were generated using a TA.XT*plus* texture analyzer (Stable Micro Systems, Hamilton, MA) that measures force and acoustic properties of the fruit. A detailed description of the instrumental methods was described in Chang et al. [27]. The selection of the individuals was based on 3-years of instrumental data (2016–2018). Ten fruit of each progeny individual were harvested each year.

Fruit were peeled, and cortex samples were collected at two time points, at harvest and after 8-week cold storage (0  ±  1 °C and 95% relative humidity). Diced fruit pieces from individual fruit were packaged in aluminum foil and frozen with liquid nitrogen, then kept at −80 °C for later RNA extraction.

### 5.2. RNA Sample Preparation and RNA Sequencing

RNA was extracted from 10 g of cortex tissue from each fruit. An RNA extraction method developed by Lopez-Gomez and Gomez-Lim [69] for fruit with high polysaccharides was used with modification [21,22]. The extracted RNA was further purified with the RNeasy Midi kit (Qiagen, Valencia, CA) following manufacturer protocols. DNase I (New England Biolabs, Ipswich, MA) was applied to eliminate genomic DNA contamination in the RNA extracts.

RNA-Seq of samples harvested in 2018 was performed at the University of Minnesota Genomic Center. The RNA integrity and concentration were measured with the RNA ScreenTape System (Agilent Technologies, Santa Clara, CA) and RiboGreen RNA quantification kit (Molecular Probes, Eugene, OR). The concentrations of the RNA samples were from 40 to 100 ng/μL, with RNA integrity number (RIN) larger than eight. cDNA libraries were constructed using Illumina TruSeq RNA sample preparation protocol. The parent samples were sequenced in 2014 on an Illumina HiSeq 2000 platform with eight libraries pooled into one lane of an Illumina flowcell. The progeny samples were sequenced in 2018 on an Illumina HiSeq 2500 platform with 12 libraries pooled in one lane. A total number of 60 paired-end RNA-Seq datasets were generated including the two parents and the eight individuals sampled at two time-points (at harvest and after storage) with three biological replicates (i.e., three fruit from the same tree) of each individual and time-point (Appendix A and Appendix A).

Quality control of the sequence reads was performed using FastQC [70] (version 0.11.9). Adapter contamination and low-quality reads (with Phred quality score < 30) were filtered using Trimmomatic [71] (version 0.39). The filtered reads were mapped to the apple reference genome GDDH13 v1.1 [72] using HISAT [73] (version 2.1.0). SAMTools [74] (version 1.9) was used to sort the aligned reads based on their locations in the genome, while featureCounts [75] (version 1.5.2) was used to count the number of the sorted reads as the expression level of each gene. The results of RNA sequencing and read alignments are shown in the Appendix A (Appendix A).

### 5.3. Differential Expression Analysis

Statistical analyses of gene expression were performed using edgeR software [76] (version 3.11). Genes with low counts across all datasets were removed using default settings in the “filterByExpr” function. The trimmed mean of M-value (TMM) method was applied for data normalization using the “calNormFactor” function to reduce technical variations. A multidimensional scaling (MDS) plot was generated from the top 500 genes with the largest standard deviation between samples using the limma R package [77] (version 3.11) to examine the relationships among samples.

Gene expression was normalized to counts per million mapped reads (CPM). The significance level was set at FDR < 0.05, and genes with log2-fold difference > 1 and CPM values > 1 were considered as differentially expressed. Three comparisons were made: (1) ’Honeycrisp’ vs. MN1764, (2) “Retain” vs. “Lose”, and (3) “Retain” vs. “Non-crisp” (Appendix A). To identify differentially expressed genes (DEGs) that commonly occurred in all the comparisons, a Venn diagram was generated using the “vennDiagram” function in the limma R package. Heatmaps visualizing the expression patterns of the DEGs was generated using the pheatmap R package [78].

Gene functional annotation was obtained from the Genome Database of Rosaceae Species (GDR, https://www.rosaceae.org/species/malus/all). Gene Ontology terms for each gene were assigned using the PANNZER2 webservice with default settings [79]. GO enrichment analyses were conducted using goseq software [80] (version 1.40.0), which was specifically designed to minimize length-derived bias that may affect RNA-Seq data. GO terms with FDR < 0.05 were considered as significantly enriched.

### 5.4. Gene Validation Using NanoString nCounter^®^ and qRT-PCR

A subset of the DEGs that showed > 1.5 log2-fold difference were further validated using the nCounter^®^ analysis system (NanoString Technology, Seattle, WA) with a customized CodeSet designed and created to target the 47 genes of interest (Appendix A). Three housekeeping genes including casein kinase 1 isoform delta like (*CKL*, MD09G119011), type 1 membrane protein like (*TMp1*, MD04G1005300), and dihydrolipoamide dehydrogenase (*DLD*, MD16G1145800) that were consistently expressed in apple fruit over storage were selected for normalizing expression of the genes of interest [81]. To evaluate the consistency of the genes, RNA samples from two different harvest years were included in this experiment. A total of 96 samples consisting of the two parents and six progeny individuals (three each from the “Retain” and “Lose” individuals), each collected at two time-points (at harvest and after two months of cold storage) and from two years (2018 and 2019) with three biological replicates each, were analyzed. The nCounter^®^ data was adjusted using the manufacturer-provided spiked positive and negative controls.

Eight genes, including two cell wall-related genes and six genes that showed inconsistent expression between RNA-Seq and nCounter^®^ results, were further examined using qRT-PCR. *DLD* (*MD16G1145800*) was selected as the housekeeping gene used to compare to genes of interest that were tested. The primers for the genes were designed (Appendix A) using the Integrated DNA Technologies (IDT) online tool (https://www.idtdna.com). Reverse transcription reactions were performed using GoScript^TM^ Reverse Transcriptase (Promega, Madison, WI) following manufacturer protocols using Oligo(dT)_15_ and random primers. Real-time PCR was performed on a CFX96^TM^ thermal cycler (Bio-Rad, Hercules, CA), with SYBR® Green Supermix (Bio-Rad, Hercules, CA) as the fluorescence reagent. Reactions for the target and housekeeping genes were performed in duplicate with a total volume of 20 μL. PCR was conducted with the following conditions: initial incubation at 95 °C for 5 min, followed by 40 cycles of denaturation at 95 °C for 20 s, annealing at 60 °C for 30 s, extension at 72 °C for 30 s, and finishing with 72 °C for 5 min. Gene expression levels were calculated using the 2^−ΔΔCt^ method [82].

## Figures and Tables

**Figure 1 plants-09-01335-f001:**
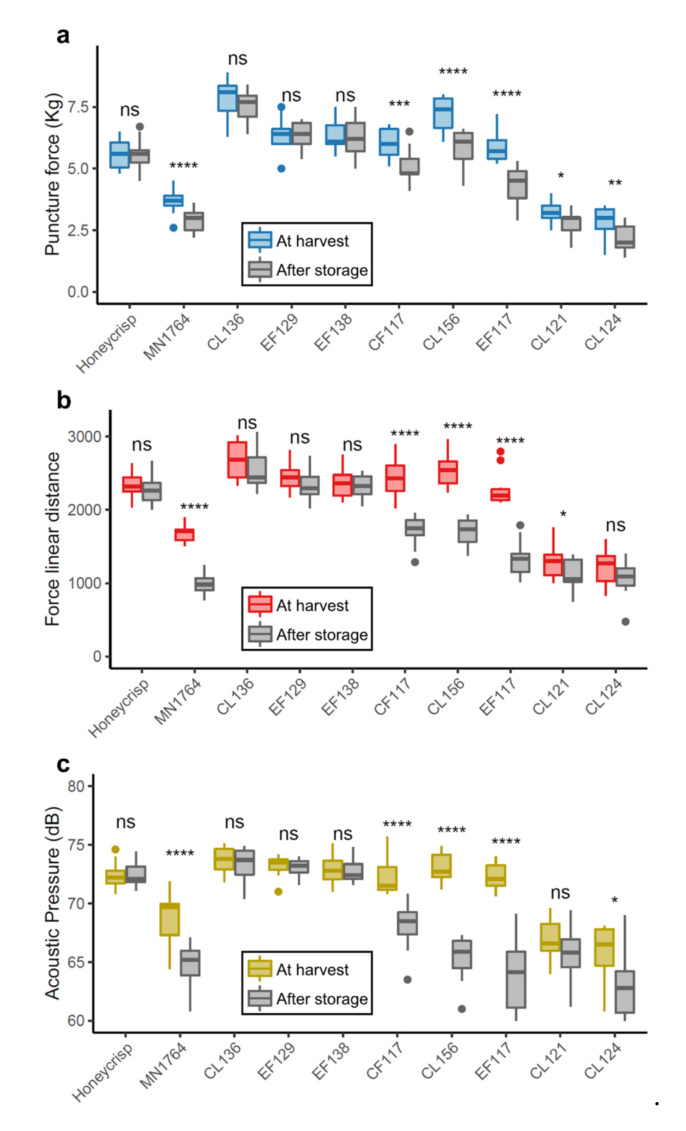
Three instrumental texture measures, including (**a**) puncture force, (**b**) force linear distance, and (**c**) acoustic pressure, of the parents (Honeycrisp and MN1764) and the progeny individuals at harvest and after 2-month cold storage. The results were obtained from three-year measurements (2016–2018). For each time point, five fruit were sampled from each parent and individual. The symbols indicate statistical significances between fresh and stored fruit of an individual: ns = not significant, * *p* < 0.05, ** *p* < 0.01, *** *p* < 0.001, **** *p* < 0.0001.

**Figure 2 plants-09-01335-f002:**
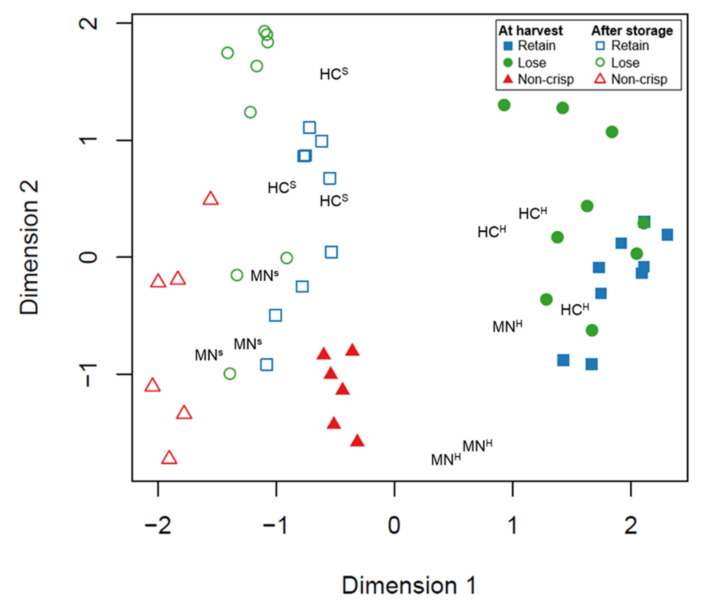
Multidimensional scaling (MDS) plot based on the expression levels of the top 500 most divergent genes. The distance between each pair of samples is the root-mean-square deviation for the top genes. HC^H^ = ’Honeycrisp’ at harvest, HC^S^ = ’Honeycrisp’ after 8-week cold storage, MN^H^ = MN1764 at harvest, MN^S^ = MN1764 after 8-week cold storage. Each symbol represents a replicate sample.

**Figure 3 plants-09-01335-f003:**
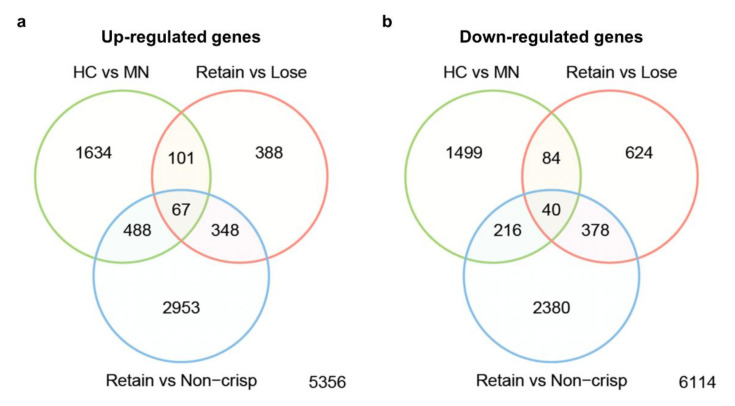
Venn diagram showing the numbers of differentially expressed genes (DEGs) commonly identified between the three comparisons. (**a**) Up-regulated gene = DEGs highly expressed in ’Honeycrisp’ (HC) and “Retain” group, and (**b**) down-regulated genes = DEGs highly expressed in MN1764 (MN), “Lose”, and “Non-crisp” groups.

**Figure 4 plants-09-01335-f004:**
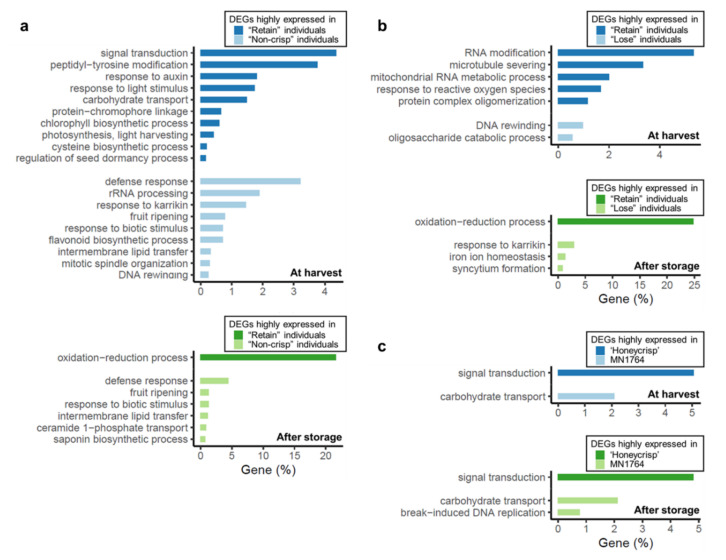
Enriched Gene Ontology (GO) terms associated with the differentially expressed genes (DEGs) distinguished between the (**a**) “Retain” and “Non-crisp” groups, (**b**) “Retain” and “Lose” groups, and (**c**) ’Honeycrisp’ and MN1764 at harvest and after 2-month storage. A false discovery rate (FDR) < 0.05 was used as the threshold for identifying significantly enriched GO terms.

**Figure 5 plants-09-01335-f005:**
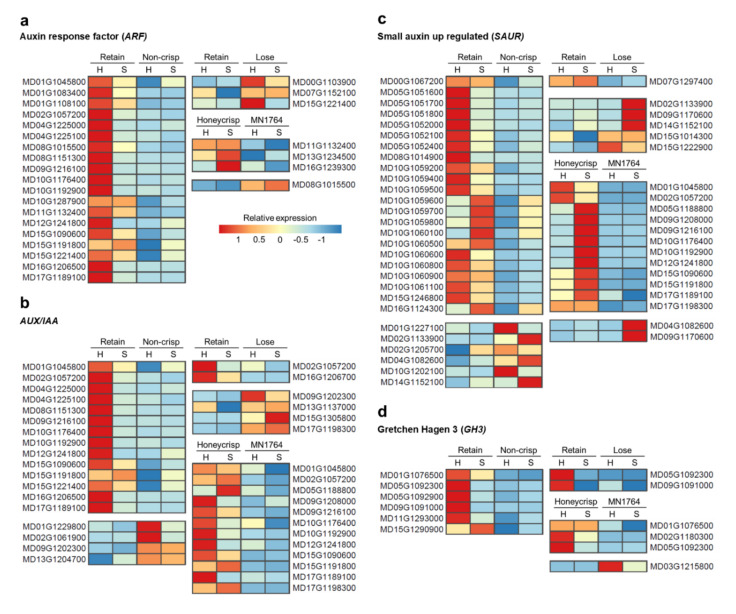
The expression patterns of the differentially expressed genes (DEGs) involved in the auxin-activated signaling pathway. Two genes, (**a**) *ARF* and (**b**) *AUX/IAA,* that are associated with auxin signaling, and two genes, (**c**) *SAUR*, and (**d**) *GH3,* that are associated with auxin response were studied. The relative expression is the ratio of gene expression compared to the average. H = at harvest, and S = after 8-week cold storage.

**Figure 6 plants-09-01335-f006:**
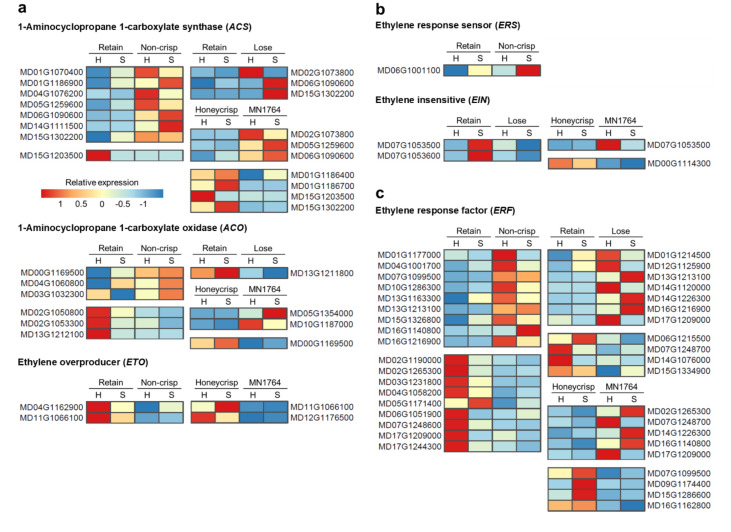
The expression patterns of the differentially expressed genes (DEGs) involved in (**a**) ethylene biosynthesis, (**b**) signaling, and (**c**) response. The relative expression is the ratio of gene expression compared to the average. H = at harvest, and S = after 8-week cold storage.

**Table 1 plants-09-01335-t001:** Differentially expressed genes (DEGs) with functions associated with cell wall synthesis or modification identified at harvest, or after 2-month cold storage. The expression levels of selected cell wall-genes were compared between (**a**) “Retain” vs. “Non-crisp”, (**b**) “Retain” vs. “Lose”, and (**c**) ’Honeycrisp’ vs. MN1764. The Gene IDs listed in the table were DEGs highly expressed in the corresponding phenotypic group. - = no DEGs.

**(a)**	**At harvest**		**After Storage**	
	**Retain**	**Non-crisp**	**Retain**	**Non-crisp**
α-Arabinoglucosidase	-	MD08G1221800 MD16G1158300	-	-
β-Galactosidase	MD08G1023600 MD15G1251100	MD00G1018700 MD02G1079200	MD15G1251100	-
Cellulose synthase/Cellulose synthase- like	MD01G1236500 MD01G1236600 MD02G1311600 MD03G1133700 MD07G1309200 MD08G1147200 MD10G1029800 MD15G1123200 MD15G1340200 MD17G1099800	MD03G1028900 MD03G1178600 MD13G1209200 MD15G1150300	MD01G1236500 MD01G1236600 MD08G1076900 MD08G1126200 MD09G1037900 MD15G1064400 MD15G1415100 MD15G1415200 MD17G1144800	MD03G1028900 MD08G1147200
Expansin	MD00G1125400 MD01G1166700 MD04G1129800 MD07G1233100 MD09G1279500	MD03G1090700 MD05G1130300 MD06G1041000 MD06G1195100	MD00G1125400 MD01G1166700 MD04G1052600	-
Galacturonosyltransferase	MD04G1181600 MD05G1363900 MD09G1041100 MD09G1061900 MD11G1318000 MD13G1084900 MD17G1141200 MD16G1084000	-	MD09G1041100 MD10G1140000 MD13G1084900 MD16G1084000 MD17G1141200	-
Pectate lyase	MD14G1167100	-	-	MD01G1100600
Pectin methylesterase	MD00G1105300 MD04G1198000 MD07G1289200 MD12G1198000 MD15G1222000	MD08G1195600	MD02G1104600 MD13G1149800 MD15G1222000	MD06G1064700 MD07G1255000 MD08G1195600 MD09G1054900 MD16G1150200
Polygalacturonase	MD01G1068900 MD01G1069000 MD06G1105300 MD09G1290500 MD12G1064100 MD16G1161800	MD00G1140300 MD03G1162500 MD07G1279000 MD09G1030100 MD09G1030200 MD10G1179100	MD00G1140300 MD06G1105300	-
Xyloglucan endotransglucosylase/ hydrolase	MD02G1192600 MD09G1102600 MD10G1315100 MD15G1303500 MD16G1014000 MD16G1091200	MD16G1145200 MD17G1140000	MD04G1020100 MD15G1303500 MD16G1091200	MD13G1237300 MD16G1278900
**(b)**	**At harvest**		**After Storage**	
	**Retain**	**Lose**	**Retain**	**Lose**
α-Arabinoglucosidase	-	-	-	MD16G1158300
β-Galactosidase	-	MD00G1018700 MD02G1079200	-	-
Cellulose synthase/ Cellulose synthase- like	MD05G1296600 MD13G1209200	MD03G1028900 MD15G1415100 MD15G1415200 MD16G1145200 MD17G1038900	MD05G1296600 MD13G1209200	MD03G1028900 MD17G1099500 MD17G1099600
Expansin	MD05G1130300	MD11G1054500 MD16G1070600	MD05G1130300 MD17G1271500	MD01G1166700 MD06G1195100 MD07G1233100 MD13G1070200
Galacturonosyltransferase	-	-	MD17G1141200	-
Pectate lyase	-	-	-	MD06G1161400
Pectin methylesterase	MD06G1191000	MD13G1149800	-	MD08G1195600
Polygalacturonase	MD06G1105300	MD10G1179100	MD13G1092000	MD00G1087900 MD09G1030100 MD09G1030200
Xyloglucan endotransglucosylase/ hydrolase	MD10G1315100 MD13G1237300 MD15G1303500 MD16G1091200	MD16G1145200	MD15G1303500 MD16G1091200	MD13G1268900 MD16G1145200
**(c)**	**At harvest**		**After Storage**	
	**Honeycrisp**	**MN1764**	**Honeycrisp**	**MN1764**
α-Arabinoglucosidase	-	-	-	MD08G1221800 MD16G1158300
β-Galactosidase	-	-	MD08G1139000 MD09G1192500	MD11G1133400
Cellulose synthase/ Cellulose synthase- like	MD03G1029100 MD03G1178600 MD04G1173700 MD15G1340200	MD03G1029000 MD03G1133700 MD15G1415100 MD15G1415200	MD03G1029100 MD03G1178600 MD04G1173700 MD11G1156200 MD13G1209200 MD15G1340200 MD17G1099600	MD03G1133700
Expansin	-	MD16G1070600	-	MD04G1052600 MD10G1133200
Galacturonosyltransferase	MD10G1140000 MD17G1141200	MD00G1136600 MD04G1181600	MD09G1093100 MD10G1140000 MD11G1318000	-
Pectate lyase	-	MD05G1179500	-	-
Pectin methylesterase	MD01G1220700 MD06G1191000 MD09G1172600	MD06G1191000 MD08G1195600 MD11G1307500	MD01G1220700 MD09G1172600	MD11G1307500 MD16G1150200
Polygalacturonase	MD03G1292400 MD15G1441700	-	-	MD07G1279000 MD10G1179100
Xyloglucan endotransglucosylase/ hydrolase	MD10G1315100 MD16G1091200	-	MD16G1091200	MD09G1152600 MD09G1152700 MD13G1237300 MD17G1140000

**Table 2 plants-09-01335-t002:** The numbers of gene counts in the fruit samples at harvest and after 8-week cold storage measured using NanoString nCounter^®^ technology. (**a**) Genes that were highly expressed in ’Honeycrisp’ (HC) and/or the “Retain” group fruit, and (**b**) genes that were highly expressed in MN1764 (MN) and/or “Lose” group fruit. The genes that were differentially expressed in both parent and progeny samples were the primary candidates, while the genes that were only differentially expressed in the parent or the progeny samples were the secondary candidates.

**(a) Gene ID**	**At harvest**			**After storage**			**At harvest**			**After storage**			**Gene function**
	**HC**	**MN**	**Diff. ^1^**	**HC**	**MN**	**Diff.**	**Retain**	**Lose**	**Diff.**	**Retain**	**Lose**	**Diff.**	
***Primary candidate gene***													
MD01G1062800	6052	5851	0.0 ^NS^	6731	2135	−1.7 **	9720	5414	−0.8 **	4488	5713	0.3 ^NS^	*PIP1* | aquaporin
MD03G1019900	14	3	−2.1 *	9	3	−1.5 *	27	13	−1.1 **	11	6	−0.8 **	*RLK1* | receptor-like protein kinase
MD05G1092300	128	24	−2.4 *	22	6	−1.9 ^NS^	101	9	−3.4 **	12	8	−0.6 ^NS^	*GH3* | auxin-responsive protein
MD07G1237500	659	44	−3.9 **	909	12	−6.3 **	600	253	−1.2 **	269	149	−0.9 **	*RPL* | ribosomal protein
MD07G1247100	561	179	−1.7 **	545	154	−1.8 **	448	274	−0.7 **	377	293	−0.4 *	*PMSR* | peptide met S-oxide reductase
MD07G1259200	132	3	−5.3 **	138	15	−3.2 **	141	62	−1.2 **	145	84	−0.8 **	*RPM* | disease resistance protein
MD07G1270800	190	27	−2.8 *	40	23	−0.8 ^NS^	209	129	−0.7 **	8	11	0.4 ^NS^	*TUB* | tubulin
MD07G1274100	12	2	−3.0 **	9	2	−2.1 **	8	5	−0.8 **	17	9	−0.9 **	*SK* | SKP1-like protein
MD08G1106600	109	6	−4.2 **	24	15	−0.7 ^NS^	58	34	−0.8 **	16	20	0.3 ^NS^	*scpl* | serine carboxypeptidase
MD14G1056600	107	38	−1.5 **	173	139	−0.3 ^NS^	132	39	−1.8 **	253	107	−1.2 **	function unknown
MD14G1110100	82	2	−5.4 **	57	4	−3.7 **	66	26	−1.3 **	82	19	−2.1 **	function unknown
MD15G1297000	91	10	−3.3 **	80	6	−3.7 **	123	25	−2.3 **	80	24	−1.7 **	*APK* | adenylyl-sulfate kinase
MD16G1091200	274	12	−4.5 **	83	7	−3.5 **	291	36	−3.0 **	90	22	−2.0 **	*XTH* | xyloglucan endotransglucosylase
***Secondary candidate gene***													
MD01G1042500	4	11	1.4 ^NS^	7	8	0.2 ^NS^	2	6	1.4 **	4	18	2.1 **	*ELI* | elicitor-activated gene
MD02G1057200	12	16	0.4 ^NS^	17	7	−1.3 *	8	7	−0.2 ^NS^	9	8	−0.2 ^NS^	*AUX/IAA* | auxin-responsive protein
MD05G1098700	367	111	−1.7 *	490	357	−0.5 ^NS^	140	138	0.0 ^NS^	288	317	0.1 ^NS^	*LACS* | AMP-dependent synthetase
MD10G1315100	155	76	−1.0 ^NS^	23	9	−1.3 *	101	120	0.2 ^NS^	16	19	0.2 ^NS^	*XTH* | xyloglucan endotransglucosylase
MD11G1230200	38	13	−1.5 **	31	18	−0.7 *	38	32	−0.3 ^NS^	71	56	−0.3 ^NS^	function unknown
MD15G1203500	2823	77	−5.2 *	179	33	−2.4 ^NS^	305	531	0.8 ^NS^	5	15	1.6 **	*ACS* | ACC synthase
MD17G1141200	92	32	−1.5 **	110	43	−1.4 *	88	75	−0.2 ^NS^	80	55	−0.6 ^NS^	*GAUT* | galacturonosyltransferase
**(b) Gene ID**	**At harvest**			**After storage**			**At harvest**			**After storage**			**Gene function**
	**HC**	**MN**	**Diff.**	**HC**	**MN**	**Diff.**	**Retain**	**Lose**	**Diff.**	**Retain**	**Lose**	**Diff.**	
***Primary candidate gene***													
MD00G1036800	52	123	1.2 ^NS^	24	96	2.0 *	44	172	2.0 **	45	92	1.0 **	*ABCG* | ABC transporter G
MD01G1213100	789	1587	1.02 ^NS^	2193	7274	1.7 **	859	1041	0.3 ^NS^	2259	11492	2.3 **	*CHIA* | chitinase
MD03G1108400	62	194	1.6 **	124	819	2.7 **	12	37	1.6 **	134	444	1.7 **	*GLTP* | glycolipid transfer protein
MD05G1310400	17	2357	7.1 **	23	1585	6.1 **	391	2023	2.4 **	33	769	4.5 **	protein E6-like
MD05G1313300	10	5	−1.0 *	15	80	2.4 *	4	8	0.9 **	11	60	2.5 **	function unknown
MD06G1233800	51	146	1.52 ^NS^	78	153	1.0 *	90	155	0.8 **	89	228	1.4 **	monoacylglycerol lipase-like
MD08G1127900	8	34	2.02 ^NS^	15	50	1.7 **	5	23	2.3 **	10	65	2.8 **	*AFR* | F-box protein AFR-like
MD10G1179100	3074	8577	1.52 ^NS^	1602	64580	5.3 **	308	7092	4.5 **	42086	76687	0.9 **	*PG* | polygalacturonase
MD11G1189000	460	734	0.72 ^NS^	1626	3551	1.1 *	272	922	1.8 **	662	10298	4.0 **	*BG* | glucan endo−1,3-β-glucosidase
MD12G1164900	41	208	2.32 ^NS^	22	161	2.9 **	58	185	1.7 **	30	84	1.5 **	PPR | pentatricopeptide repeat protein
MD12G1183000	46	66	0.52 ^NS^	80	250	1.6 **	35	45	0.3 ^NS^	77	138	0.8 **	LURP-one-related 15-like
MD13G1112700	16	8	−1.0 ^NS^	9	36	2.0 **	12	26	1.1 **	17	27	0.7 *	*CYP* | cytochrome P450
MD15G1023600	896	817	−0.1 ^NS^	1433	13417	3.2 *	89	500	2.5 **	2485	7947	1.7 **	*JMT* | jasmonate O-methyltransferase
***Secondary candidate gene***													
MD03G1060100	72	58	−0.3 ^NS^	44	122	1.5 ^NS^	26	41	0.6 *	27	83	1.6 **	*LBD* | LOB domain-containing protein
MD05G1297900	11	40	1.9 ^NS^	11	172	4.0 **	10	39	1.9 ^NS^	147	140	−0.1 ^NS^	*EFR* | EF-TU receptor
MD05G1349800	1837	1358	−0.4 ^NS^	2045	1726	−0.2 ^NS^	1434	2284	0.7 **	1382	2055	0.6 **	*WRKY* | WRKY transcription factor
MD06G1090600	14	45	1.7 ^NS^	46	112	1.3 ^NS^	5	10	1.1 ^NS^	76	507	2.7 **	*ACS* | ACC synthase
MD16G1158300	2231	5769	1.4 ^NS^	13467	23748	0.8 ^NS^	5066	7463	0.6 ^NS^	19752	27723	0.5 *	*α -AF* | α -arabinofuranosidase
MD16G1277800	3	13	2.1 ^NS^	5	10	0.9 ^NS^	3	10	1.9 **	10	17	0.8 **	*NRT* | nitrate transporter
MD17G1256100	19	56	1.5 ^NS^	34	67	1.0 ^NS^	27	42	0.6 ^NS^	81	286	1.8 **	*SFBB* | F-box family protein

^NS^ not significant, * *p* < 0.05, ** *p* < 0.01; ^1^ Diff. = Log_2_ fold-diffe.

## Data Availability

The raw sequence data were deposited into NCBI Sequence Read Archive and can be accessed with the accession number PRJNA645625.

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
