# Peer review of "Identification of Candidate Genes Involved in Fruit Ripening and Crispness Retention Through Transcriptome Analyses of a ‘Honeycrisp’ Population"

_plants, 2020, doi:10.3390/plants9101335_

Round 1

Reviewer 1 Report

This is an interesting and well-written article, studying genes associated with texture retention in Honeycrisp apple. The text is rather long, but a substantial amount of work is described and appears to be based on thorough foundations. Of course, gene candidates identified by these population studies are at this point just correlative in nature, but the work has thrown up a range of potentially useful candidates for further research in auxin, ethylene and cell wall metabolism.

Some minor comments

It is not clear how eight progeny individuals from each of three groups (i.e., 24 individuals, lines 433-4) was reduced to eight individuals total (line 82 and Fig. 1).

In Fig. 3, how was it determined if a DEG was ‘highly expressed’?

Fig. 6a, should be carboxylate, not carboylate (two instances).

Lines 218-9, the heading 2.5 has lost its italics and line spacing.

In Table 1, it is not identified which of the DEGs are up and which are down in the different phenotypes. This would be of some interest, and I wonder if color coding could be used to show it.

Lines 279-289. I struggled with the logic in this paragraph. Assuming the “Lose” fruit lost crispness due to ethylene, is this consistent with the final statement that the “Lose” fruit had higher auxin so lower ethylene and slower ripening.

Line 374, I couldn’t find a mention of transgenic PG1 apples in reference 23.

Line 381, AFase is not arabinoglucosidase.

Lines 412-4, the transgenic data of Miedes et al. (2010, J. Agric. Food Chem. 58:5708-5713) also point towards a role for at least some XET activity in maintaining, rather than degrading, the wall structure.

Line 443, was a biological replicate a single fruit, or several pooled fruit, and if so was it from more than one of the 2-4 replicate trees?

Line 473 says gene counts were in CPM, yet line 222 says they were FPKM.

A quick skim through the reference list found several entries that need checking, lacking volume or pages/article number, incorrect journal name, lacking publisher, e.g., refs 12, 29, 35, 36, 41, 54, 55, 56, 57, 59, 66, 79.

Reviewer 2 Report

The authors aim to understand and reveal the molecular mechanisms of apple fruit crispiness retention, an exceptional postharvest trait found in Honeycrisp apples. Transcriptomes have been made from three apple crispiness lines: apples retaining crispiness during postharvest storage, apples loosing crispiness during postharvest storage and apples, which do not show crispiness. The manuscript uses an appropriate approach to use transcriptomics to reveal the genes responsible for crispiness retain. The manuscript is well written, however, there are some shortages in the manuscript, and modifications are needed before it is ready for publication.

  • The title indicates that novel genes have been found in this study and abstract indicates that functional analyses have been performed for these novel genes to reveal their function. However, the novelty of the identified genes remains unclear. The described genes were identified by annotation as genes with known functions in ethylene, auxin, and cell wall metabolism. In my opinion, the authors should indicate more clearly which genes are novel (not described earlier) in abstract and in conclusions, and add the data from functional analyses to support the claim of their novel function, or otherwise omit the words ‘novel genes’ from the title.
  • In the abstract, authors indicate that functional analyses have been performed. However, no “functional analyses” have been presented but the term has been used to describe functional annotation and gene expression analyses. I suggest to make corrections to the abstract by replacing “functional analyses” to “functional annotation”, also thorough the manuscript.
  • The MN1764 line has not been explained. I suggest to add explanation of this apple line in introduction section with references.
  • The quality of transcriptomics data (including average length of unigenes) have not been presented and should be added to be able to evaluate the quality and reliability of the constructed libraries.
  • The amount of identified DEGs is surprisingly high (4129; 2030; 6870) and the authors should comment on this and their decision for the threshold value. Using higher threshold would possible give to better overview of the real differences; the genes that are truly highly up-regulated/down-regulated when softening begins and could lead to identification of novel regulators.
  • There are high amount of genes of same function identified, such as ARF, ERF and ACS. The authors do not comment if they are sequences from the same gene or representing different genes of large gene family. This should be clarified.
  • The authors have concentrated on genes related to ethylene and auxin metabolism, but in recent years plant hormone ABA has also been shown to have a role in apple fruit ripening. I suggest to add analysis/comment of ABA genes to the manuscript.
  • There are some lacking in the gene expression data: The expression estimated concerning genes related to auxin and ethylene metabolism are presented in figures 5 and 6. However, the estimated expressions concerning cell wall-related genes have been placed on tables and the presenting style is not as clear.
  • Authors indicate that the gene expression estimated from RNA-Seq data differentiates from the qRT-PCR data. The authors have placed the qRT-PCR data in the Supplementary files although they indicate that it would probably be more reliable data. I would also assume that more reliable overview of the expression change would have been obtained by following the expression over the ripening and storage with more than two time-points.
  • There are lacking in the presentation of Figures: 1) Figure 1 and its legend needs corrections: there seems to be two symbols representing p<0.01, and I suggest to delete the other. There is also a lack in information of the statistical analysis: “The symbols indicate statistical significances” compared to what? In my opinion, ANOVA should be used to compare significant changes between the lines. I think this was the purpose of the measurements. 2) The Figure 2 is somewhat confusing and fails to indicate the line of each sample indicated in the figure. This should be clarified. 3) Figure 3: The difference between 3a and 3b figures is not clear due to similar labeling in both figures. It would be more clear to discuss a) up-regulated and b) down-regulated genes, or a) HC vs MN and b) MN vs HC. This should be clarified.
  • The conclusion-section lacks the discussion of the novelty and importance of the work as the outcome of the study as hinted in  the title. Also, the regulatory mechanism that was intended to reveal remains unclear. I think it would be easy to draw such a summary figure to the manuscript. Therefore, I suggest to rewrite the conclusions section concentrating on the novelty of the work, and place the conclusion after discussion section.

Reviewer 3 Report

Dear Author,

Nice work, just a minor, minor comments.

Round 2

Reviewer 2 Report

The authors aim to understand the molecular mechanisms of apple fruit crispiness retention, an exceptional postharvest trait found in Honeycrisp apples. Transcriptomes have been made from three apple crispiness lines: apples retaining crispiness during postharvest storage, apples loosing crispiness during postharvest storage and apples, which do not show crispiness. The manuscript uses an appropriate approach to reveal the genes responsible for crispiness retain. The manuscript is well written. The issues raised in revision have been answered/corrected in satisfactory level to enable its publication.